# Integration of residents' experiences into economic planning process of coastal villages: Evidence from the Greater Hangzhou Bay Rim Area

**Xinkai Wang**[1,2], **Zhirong Wang**[1]*, **Yao Wang**[1‡], **Qianqian Zhang**[1‡], **Tengyue Zhang**[1‡], **Jia Yao**[1‡], on behalf of The Rural Development and Ecological Technology Research Institute¶

**1** School of Design, Ningbo Tech University, Ningbo, Zhejiang, People's Republic of China, **2** Department of Geography and Planning, University of Liverpool, Liverpool, United Kingdom

☯ These authors contributed equally to this work.
‡ YW, QZ, TZ and JY also contributed equally to this work.
¶ Membership of the Rural Development and Ecological Technology Research Institute is provided in the Acknowledgments.
* wzhirong@nit.net.cn

**Data Availability Statement:** All relevant data are within the manuscript and its Supporting Information files.

## Abstract

Public value is gaining prominence from both academics and politicians with regards to China's rural development. However, rural planning authorities and practitioners showed limited confidence on public, which manifests as few public perceptions were integrated into the planning documents. This study explores the potential role of residents' experiences in illustrating local economic development within the context of coastal villages in which economic and industries are rapidly transforming. Two case studies from within the locale of the Greater Hangzhou Bay Rim Area are used in this article to examine the gap between residents' experiences and the actual economic development that has occurred. The main findings suggest that rural residents can directly reflect upon both current and historic trends of local economic development. Moreover, household income satisfaction (HIS) is a comprehensive notion of residents' experiences, and indicates social and economic sustainability of industrial transformation, or "thriving business", that have been highlighted in coastal villages. Public experiences could therefore act as a valid and accessible evidence for planners in rural economic planning in China and other developing countries.

## Introduction

The sustainable agenda of rural areas has gained prominence in politics and academia since a coherent package of policies on rural development were unveiled by the China's central government [1]. Moreover, increasing attention has been paid to the environmental dimension of rural China, as well as to socio-economic aspects, including social capital, social justice, public supervision, participatory governance, and residents' experiences [2–5]. In 2019, the relevance

**Funding:** This research was funded by the following grants: Zhejiang University Ningbo Institute of Technology, 20190418Z0047, Dr. Xinkai Wang. Zhejiang University Ningbo Institute of Techonology, 20900545850, Dr. Xinkai Wang. Natural Science Foundation of Zhejiang Province, LQ19E080010, Qianqian Zhang. MOE (Ministry of Education in China) Project of Humanities and Social Sciences, 19YJC760122, Dr. Xinkai Wang. The funders had no role in study design, data collection and analysis, decision to publish, or preparation of the manuscript.

**Competing interests:** The authors have declared that no competing interests exist.

of residents' satisfaction in the national strategy of Rural Revitalization (a policy first stated by President Xi in 2017) was highlighted by five national ministries and commissions [6]. The residents' experiences were considered to be relevant parameter of regional and local development by politicians. Indeed, Wang and Watanabe (2019) indicated that public perceptions were significantly related to, but not consistently formed by, both perceived short-term benefits and risks in plan-making [7].

In 2018, The CPC Central Committee and the State Council stated *Strategic Planning of Rural Revitalization (2018–2022)* which highlighted the relevance of economic development in rural areas by putting "thriving business" in the first place of rural revitalization. Local decision-makers have started to deliver rural projects, especially tourism projects and in so doing, have sought to garner a greater understanding of local dynamics by engaging with the perspectives of rural residents [8, 9]. However, unlike in urban areas, the ideas of rural residents and rural public participation were not fully considered as a decisive factor in the rural planning practices. There was, or is, a concern about the participatory abilities of rural residents and the scope of the initiatives that seek to change Chinese rural society [10]. Hao (2008) even went as far as to suggest that public perceptions were emotional and fragmentary [11], and that they were also specifically and directly related to the individual environments in which residents lived. Residents' experiences were discussed as being an emotional and less rational method of evaluating regional and local development in the rural planning and design subjects [12, 13]. In addition, practitioners have, in general, shown limited confidence in the data that has been collated upon residents' experiences and how such data could be used to inform the future planning of developing coastal and inland villages in China [14, 15].

Cumulatively within these suggestions it can be seen that the relevance of residents' experiences was limited to those who were deemed to have sufficient participatory ability to make meaningful comments. As a result, data pertaining to residents' perceptions should be carefully examined to discover their potential relationship with the actual development, even if they have only a limited educational background or professional training.

To evaluate the validity of residents' experiences, especially economic experiences, in coastal rural areas, this empirical research discusses the nuances that exist between the experiential narratives of local residents and the actual development that has occurred in the individual rural areas in which they reside. In conducting a study on two cases of coastal villages in Hangzhou BRA, three research questions were proposed:

1. Do the experiences of residents reflect "thriving business" and rural economic developments;

2. which indicators best represent the economic experiences of local residents; and

3. how can this indicator and related data be managed to support the process of plan-making.

This paper also interrogates, vigorously, the potentials and possible roles of these factors in interpreting local economic development from the perspective of a diachronic and dynamic view. The following sections discuss why the study of economic experiences in rural areas could contribute to the development of rural area or other areas with similar socio-economic environment. Thereafter, the paper presented the two case studies to illustrate the role of residents' experiences in industrial transformation and changing economic environment. The discussion and conclusion sections reflect on these case studies to highlight the relevance of public value in rural China, and provides alternative solutions for future rural economic planning.

## Rural development in coastal area

Over the last two decades, coastal rural areas have, compared with inland rural areas, witnessed unique economic development that has been considered to be more dynamic with respect to urban-rural migration patterns, but also industrial transformation and income increases [16–18]. At the same time, however, the development of coastal villages has been constrained by certain issues, including a lack of ancestral ideas, village rules and regulations, and the limitations of inherent with the Chinese climate of self-government [19, 20]. In China, the traditional interpersonal relationships that exist between local residents within rural areas were named by Fei (2015) as "*Chaxu Geju*" (Patterns of Different Sequences)" [21]. This term describes the unbalanced power relationship that exists and constrains public willingness to advocate individual opinions and democratic issues in traditional rural society [22]. However, the rapid movement of urban-rural migrants into coastal rural areas has shattered the conventions of this rural relationship. In its place there is a need for the development of a bottom-up approach of rural governance and a coordination of the multiple driving forces that are exhibited by different stakeholder groups [19, 23]. In the suggestion of Li (2020), public consultation and negotiation is a useful and supplementary approach to support the planning of local and regional regeneration [24]. It follows, *ceteris paribus*, that the experience of the public and local communities should be an important parameter of rural development especially in those coastal villages that have borne witness to rapid population movement.

## Lack of a bottom-up rural economic planning

In sustainable development and planning, the evaluation of public participation should be delivered in environmental, social, and economic dimensions as a collective unit [25]. Munro et al. (2017: 9) suggests that coastal planning researchers should focus on the "breadth and representativeness of stakeholder interests" [26]. In other words, both breadth and depth of public value should be developed to cover various facets of rural development. Public value should also be analysed and synthesized based on given local contexts in order to fully explore its rationality and validity. Before identifying the representatives of stakeholder interests, the content of coastal rural planning should be discussed, explicitly. However, the engagement of local residents in rural economic planning remains variable, which has been to date discussed extensively in environmental and social facets of coastal village development.

## Environmental facets of rural development

According to *Standards for Planning of Villages (draft)* [27], rural planning focuses on the delivery of population, land use, residential buildings, infrastructure, production building, road systems, and public facilities. These seven areas can be conceptualised and clustered as representing 'the living environment and convenience'. However, this document only addresses the socio-economic implications on rural development in a limited manner. Decision-makers and planning professionals prefer to address short-term or synchronic issues in physical or living environment, such as wall colour or decoration, rather than the long-term or diachronic ones in rural planning [28]. A certain local government misinterpreted the national policy of Beautiful Countryside (a policy stated by central government of China in October, 2004), and implemented the policy through a shortcut by improving the main facade of villages [29]. In other words, practitioners lay more emphasis on the residential environment rather than developing rural communities with economic perspectives.

## Social facets of rural development

Furthermore, according to studies in developed western countries, satisfaction with regard to income is a key determinate of social sustainability. This idea has been highlighted since the seminal work of Jane Jacobs (1961) which sought to develop society based on the notion of residents' experiences and happiness within the discourse of social sustainability [30]. Also, Caprotti and Gong (2017) highlighted the challenges of living in a newly built urban area, including affordable housing, social inclusion, social equity, etc [4].

In the last two decades, there is also a certain study paid attention to the urban life of rural out-migration, i.e. housing, social differentiation, cultural recognition [31–33]. Issues include depopulation and aging were also highlighted in the rapidly industrial rural areas [34]. Compared with inland rural areas, however, there is a larger number of in-migration, especially labours and practitioners of fishery and tourism, as well as their families, in coastal villages [35]. Coastal villages started to meet challenges of in-migration brought by tourism development, industrial transformation, and counter-urbanization. This could lead a growing issue of social inclusion when an increasing number of new residents moving into or working in these specific areas.

Chinese researchers, gaining insight from the studies of western countries, attempted to illustrate the breadth of social inclusion through the lens of rearrangement, collaboration, and bifurcation of urban-rural dualism [36]. These studies achieved a consensus that psychological adaptation is considered as a further step based on the economic adaptation, including income and expenditure in rural areas.

## Economic facets of rural development

By combining existing studies and ideas, the rural development has been shown to inevitably interact with, or be impacted by the environmental, social and economic dimensions that exist within China's coastal villages. The vision of rural revitalization is to achieve comprehensive development, including prosperous industries, a liveable environment, a harmonious community, effective governance, and an affluent level of living [37]. Despite the material and ideological circumstances, the economic facets of coastal villages have, accordingly, gained growing attention from both academics and politicians [19]. According to the priorities of rural planning in Zhejiang, China (Fig 1), residential perceptions are integrated into environmental and social dimension, such as the accessibility of public transportation, living environment, protection of heritages and historic buildings, features of cultural tourism projects, and others related to the feelings and perspectives of local residents. However, decision-makers pay attention to the industrial transformation, business management, and logistics management in economic facets. These economic goals and visions are determined with a top-down manner with limited discussion as to how residents' experiences and satisfaction could continuously act as a representation of the local economic development.

## Engaging diachronic perceptions of rural residents in economic development

The experiences and perceptions of individuals has been advocated as a relevant factor for sustainable development within the discourse of urban landscape and environments, as well as with regard to regional and global climate change [4, 39–41]. It has been suggested that there are a number of practical methods of engaging with the public value in order to determine the true living environment of coastal villages. Indeed, the promotion of a satisfactory level of living environment can be seen to be closely related to the financial support and budgets

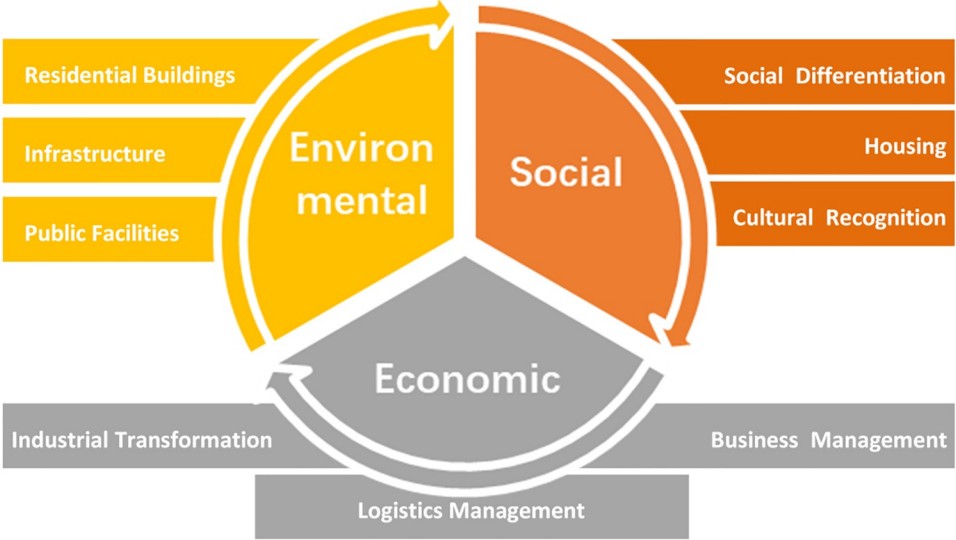

**Fig 1. Priorities of rural planning in Zhejiang, China [38] (edited by first author).**

allocated to the protection of environment and heritage in such areas as well as the level of investment that is targeted towards the provision of infrastructure and facilities. The experiences in living environment act as an indicator of rural physical environment, but there is a shortage of literatures and empirical studies indicating the potential role of residents' experiences in the analysis of economic environment in rural China.

Economic growth, according to a package of rural development policies issued by central government, has become a key factor of rural development with an increasing number of issues raised pertaining to rural out-migration, poverty, agriculture decline, and industrial depression [42–45]. It thus has been directly evaluated by highlighting and analyzing indicators that focus on various economic resources and activities. For instance, improving rural road and transport services has been argued as an important solution to stimulate economic development in rural areas [46, 47]. Unay-Gailhard and Bojnec (2019) suggested labour use and job market could be utilised to measure sustainability of economic development [48]. Household income is also regarded as an important data source by which to evaluate consumption ability and the willingness of individuals or groups in both China and developing countries to pay [49–51], as well as a being a means by which to support discourse on income inequality, poverty, and other economic issues [52, 53]. The economic facet of rural planning therefore includes both synchronic (industrial chain, logistics management, job markets, economic policy, etc.) and diachronic aspects (industrial transformation, economic growth, changing capital goods, etc.).

The existing empirical suggested that residents' experiences, especially satisfaction on household income, could be utilised as an indicator of studying aged population, tourism, social engagement, and other synchronic problems in the long-existed rural areas [48, 54]. The positive correlation between household income and satisfaction of living in rural areas has been discussed. However, it is not clear that the dynamic economic development, or a diachronic discussion of local economic environment, could be conducted based on the residents' satisfaction on household income in coastal rural areas.

This study attempts to understand the economic satisfaction, especially HIS, of two local communities in the Greater Hangzhou Bay Rim Area to address whether residents' satisfaction

can contribute to the development of the agenda of agricultural and industrial development, specifically the industrial transformation, economic growth and other diachronic aspects of economic development in coastal villages, whilst also addressing the gap in existent literature.

## Methodology

In order to investigate residents' economic experiences in coastal villages, a comparative analysis of Shamaolv (SML) and Lingang (LG) was made. Both villages are located on Gaotang Island, the second largest island in Xiangshan, Ningbo in the Greater Hangzhou Bay Rim Area (Hangzhou BRA). According to the *Yearbook of Xiangshan (2016)* [55], traditional agriculture and fisheries, the pillar of the local economy of Gaotang Island, has moved to the stage of modernization and touristization.

### Case selection

In Gaotang, there is no significant differences in population between 18 villages (Fig 2). However, residents in Jingaotang gained a growing income because Jingaotang wharf is a marked route to tourist attractions on Hua'ao Island. Moreover, Jiangbei is the central and developed village where county government located, which could catalyse economic growth in Jiangbei to some extent. To mitigate the geographical and political implications in data analysis, both Jingaotang and Jiangbei were excluded in the list of case selection.

The roadmap of the Hangzhou BRA was started through a collaboration between Shanghai municipal government and Zhejiang provincial government at the end of 2017, and engaged two mega-cities (Hangzhou and Shanghai) along with five other cities in the Yangtze River Delta (Ningbo, Shaoxing, Zhoushan, Jiaxing, and Huzhou) (Fig 3) [56]. The strategic planning of Hangzhou BRA, one of the most economically dynamic areas in China, was determined to develop regional economy by investing more than 1500 billion CNY. More than 120 projects, including ones within the energy, education, tourism, and cultural industry sectors, would be delivered to contribute to enhanced regional competitiveness and sustainability. The policy of

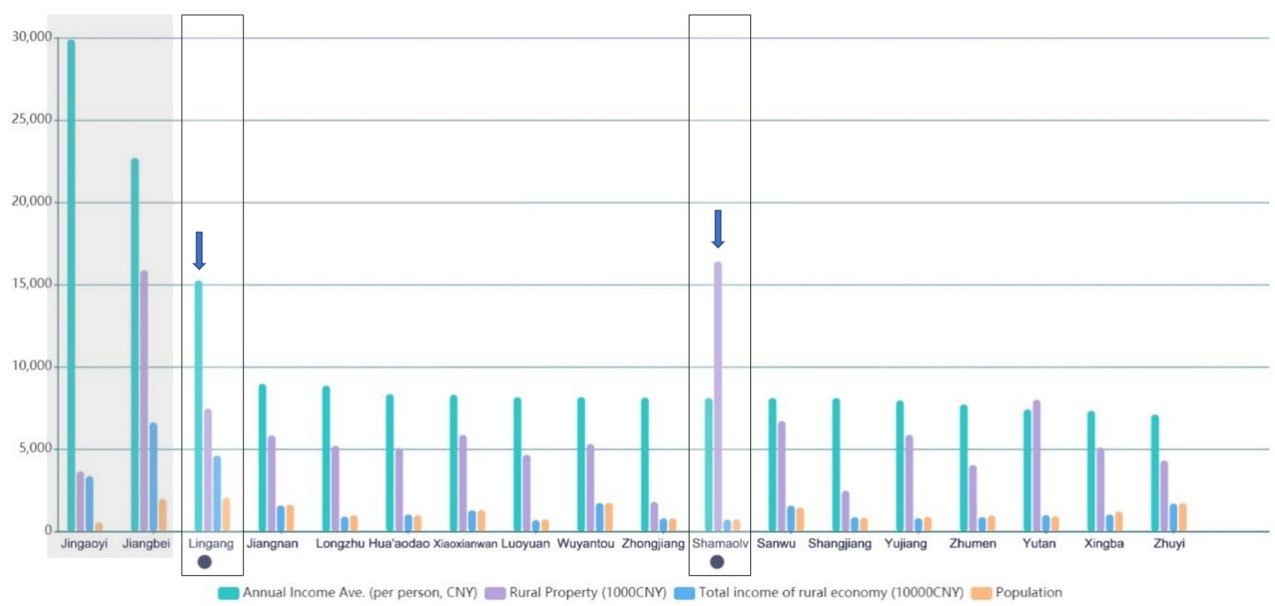

**Fig 2. Economic development in Gaotang [55] (edited by first author).**

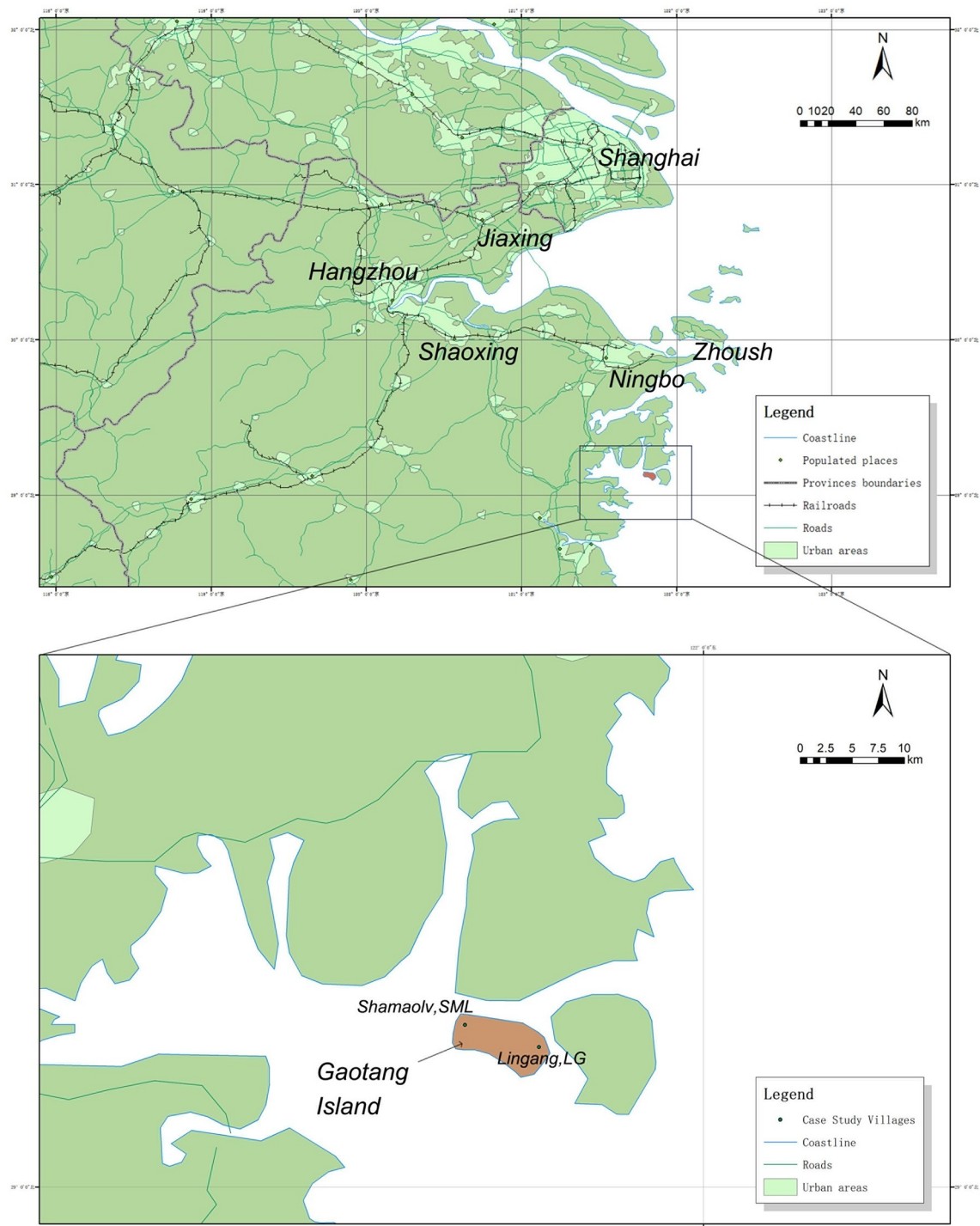

**Fig 3. SML and LG in great bay area.** a) The layout of Greater Hangzhou Bay Rim Area; b) The location of SML and LG (Reprinted from [http://www.naturalearthdata.com/] under a CC BY license, with permission from [Natural Earth (public domain)], original copyright [2020]) (edited by first author).

Hangzhou BRA inevitably impacted on the industrial and infrastructure development in Gaotang Island to some degree. Thus, the economic circumstances remain dynamic in Gaotang Island, which brings plan-makers a question as to whether the local residents' experiences could directly indicate the changing economic environment. Through so doing, practitioners would address the question as to whether rural development could contribute to the further development of the tourism and cultural industries.

Tourism and "touristization" have been extensively discussed as models of industrial transformation for fishing villages in developed countries [57, 58]. SML was reported to have established its own village company to support its industrial transformation since 2012 [59]. LG is one of biggest fishing villages in Ningbo [60]. Based on the current stages of industrial and economic development of both villages, the discussion on the economic development was conducted diachronically as a consequence of the fact that SML has transformed its fishing industry focus to tourism whilst LG remains as a fishing village.

Two villages were selected from 16 villages in Gaotang as two representatives of different stages of economic development, stage of traditional fisheries and beginning stage of tourism. According to the work of Gao (2015) in fisheries development and transformation in Zhejiang, local residents could suffer decrease on income through the industrial transformation from fisheries to tourism [61]. Moreover, the delivery of facilities and properties for developing local tourism could not directly bring local residents' income at the beginning stage.

Among the 16 listed villages, SML and LG peaked at rural property and annual income (per person) respectively (Fig 2). This indicates that SML has delivered an increasing number of properties for industrial transformation, and LG residents gained relatively higher salaries compared with other villages under current economic circumstances. SML and LG villages provide two different images of rural economic development. Consequently, SML and LG were selected as cases in this study as they act as representative vehicles by which to discuss the factors affecting the satisfaction level of local residents through a three-stage process of data collection, including questionnaire (May 2018), on-site observation (May 2018), and documentary analysis.

## Questionnaire and question design

The questionnaire was developed to investigate the relationship that exists between local economic development and the lived experiences of the public and local communities in SML and LG. Questions were designed according to the synchronic (industrial chain, logistics management, job markets, economic policy, etc.) and diachronic aspects (industrial transformation, economic growth, changing capital goods, etc.) of economic planning in rural areas as previously discussed. There were two main sections in the questionnaire. In Section 1, 13 questions (Q1-Q13) were designed to explore the economic satisfaction level of living in SML and LG by employing a standard 5-point Likert scale: 1 very unsatisfied, 2 unsatisfied, 3 moderate, 4 satisfied, and 5 very satisfied (see Table 1). The measurements were developed to identify residents' experiences and perceptions of living and working by combining the principles of data collection in rural planning as discussed above [62, 63]. Section 2 (Q14-Q18) comprised a set of questions about the identity information of interviewees, including gender, age, education background, profession, and household income. This set of questions helped the researcher to examine the viability of the data that had been collated as well as underpinning his analysis of income satisfaction.

A total of 120 households (14.07% of local households in both villages) were investigated by face-to-face questionnaire on 19th May 2018 in SML and LG. 97 (11.37% of local households in both villages) valid questionnaires were collected. The basic information of the

**Table 1. Questionnaire questions [64–66].**

|  | No. | Questions | Code |
|---|---|---|---|
| Economy | 1 | Fishery resources | FR |
|  | 2 | Fish breeding industry | FBI |
|  | 3 | Impact of Tourism development on daily life | I_TD |
|  | 4 | Local Job Opportunities | LJO |
|  | 5 | Current Working Status | CWS |
|  | 6 | Subsidy provided by government on Fishing and Farming | SPG_FF |
|  | 7 | Dividends of village company | D_VC |
|  | 8 | Household Income Satisfaction | HIS |
|  | 9 | Expectation of household income in 2018 | E_HI |
|  | 10 | Growth of household income in last 5 years | G_HI |
|  | 11 | Social Security | SS |
|  | 12 | Numbers of family members | N_FM |
|  | 13 | Number of family economic activities | N_FEA |
| Basic Information | 14 | Gender | SEX |
|  | 15 | Age | AGE |
|  | 16 | Highest Education Level | EDU |
|  | 17 | Professions | PRO |
|  | 18 | Monthly household income | MHI |

See S1 File (in English) and S2 File (in original language).

questionnaire respondents is illustrated in Table 2. To avoid repetitive respondents and mitigate against any reading problems that the participants might encounter–be it through eyesight or educational attainment level, the questionnaire questions were not sent door by door, but also translated into basic mandarin by the study investigators. The reliability and limitations of data is indicated at the end of this research.

## On-site observation and documentary analysis

To explore the in-depth reasons for residents being either satisfied or unsatisfied in SML and LG, on-site observation and documentary analysis were also employed to explore non-numeric implications, and to bring qualitative insights into the data analysis. Information and data should be understood by identifying its context and the characteristics of recipients with regard to their willingness to support rural governance and processes of decision-making [67]. Planning documents and maps were reviewed to identify the values of decision-makers and objectives of rural development for each village. This process included attaining published political documents, i.e. the Master Plan of Gaotang Island, and an unpublished report on fishing industry development provided by the Ningbo Ocean & Fishery Bureau (NOFB). The major outcome of the documentary analysis was the establishment of a baseline position from which to evaluate the gaps between the assessments of village development that were attained from those who completed the questionnaire and that contained within official planning documents.

In addition, an on-site observation of both villages was conducted in order to enable a visual assessment of the scale of implementation of each rural development to date. Tourism developments can promote landscapes and infrastructures dramatically and, in turn, directly affect the living conditions of villagers [56, 68]. The researchers conducted on-site observation as a method of data collection from which they reflected upon the reality of the rural environment and development in the two villages compared to the satisfaction levels noted by members of the public themselves.

**Table 2. Characteristics of respondents (N = 97).**

| Characteristics | N | % |
|---|---|---|
| Gender | | |
| Male | 55 | 56.7 |
| Female | 42 | 43.3 |
| Age | | |
| 14–25 | 6 | 6.2 |
| 26–35 | 12 | 12.4 |
| 36–45 | 29 | 29.9 |
| 46–60 | 36 | 37.1 |
| >60 | 14 | 14.4 |
| Education | | |
| Middle school or below | 79 | 81.4 |
| High School Diploma (or high school equivalent) | 16 | 16.5 |
| Junior college or undergraduate degrees | 2 | 2.1 |
| Postgraduate degrees or above | 0 | 0 |
| Profession | | |
| Self-employer | 27 | 27.8 |
| Fisherman or farmer | 19 | 19.6 |
| Retired | 14 | 14.4 |
| Housewives | 11 | 11.3 |
| Student | 8 | 8.2 |
| Technician or Professionals | 3 | 3.1 |
| Migrant workers | 2 | 2.1 |
| Civil servant | 1 | 1.0 |
| Enterprise Administrators | 0 | 0 |
| Not stated | 12 | 12.4 |

## Data analysis

This study adheres to the two-stage study method (see Fig 4) in order to achieve methodological triangulation. First, the questionnaire data collected in SML and LG was analysed by SPSS 19.0 separately, and then each component measurement between the two cases was compared. This provided not only an overview of living experiences in two different types of fishing villages, but also a comparative analysis of both villages from which it was possible to see how industrial transformation impact on the development of local community in SML and LG. The data analysis was also conducted through a backward design of multiple regression analysis to explore the correlation between HIS and other variables. In this way, it is possible to understand the role of modest-sized set of potential variables based on their statistical contribution [69]. Secondly, the result of the questionnaire data was merged with the observation and documents data to support the validity of data. Finally and ideally, data collected through the two-stage methods was synthesized to illustrate the key factors affecting experiences of local residents in fishing villages.

## Results: Local economy and household income satisfaction (HIS)

### Questionnaire respondents

Of 120 questionnaires distributed to potential representatives of local families in SML and LG, 97 (80.8% response rate) were valid and complete. There were more male respondents

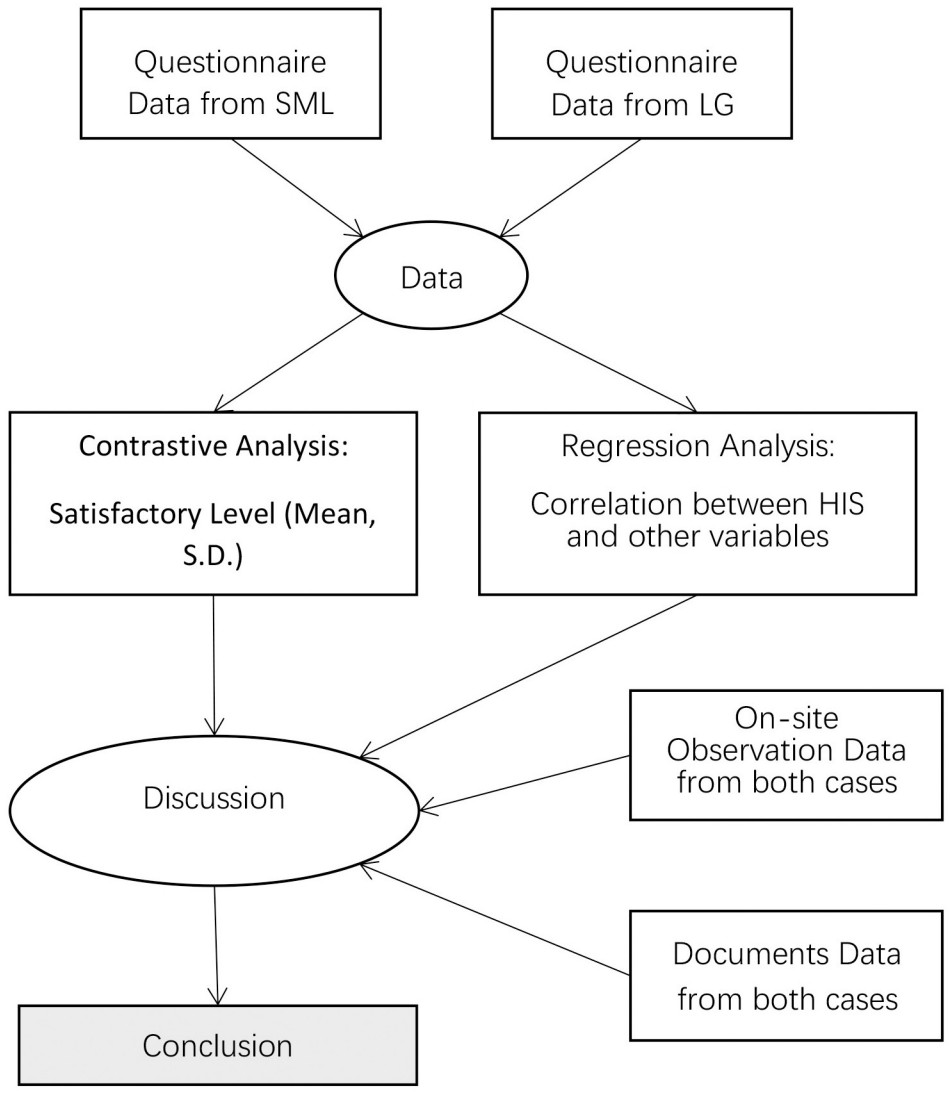

**Fig 4. Design of research methods.**

(N = 55, 56.7%) than female (N = 42, 43.3%). More than half (51.5%) of the respondents were aged above 45. The majority of participants had, at most, diploma from middle school (N = 79, 81.4%). More than one third of respondents were currently unemployed (N = 33, 33.3%), this included persons who retired, housewives, or students. 27 respondents were self-employed (27.8%) (See Table 2).

Fig 5 shows the distribution of absolute income of those who stated their household income in the two research areas. Both villages had about 40 percent of participants whose household income was more than the average level of rural areas (2365 CNY/per month [70]) in Xiang-shan, Ningbo (Fig 5a). In both LG and SML, more than 20 percent of participants achieved the average income level of town residents (4223 CNY/per month) in Xiangshan. Moreover, more than 20 percent of participants lived on government aid (612 CNY/per month) in LG. This number was reduced to about 10 percent in SML. Furthermore, the highest and second highest household income (14,583 and 9,722 CNY/per month) of participants were from SML; this village provides more flourishing career opportunities for local residents (Fig 5b). In general,

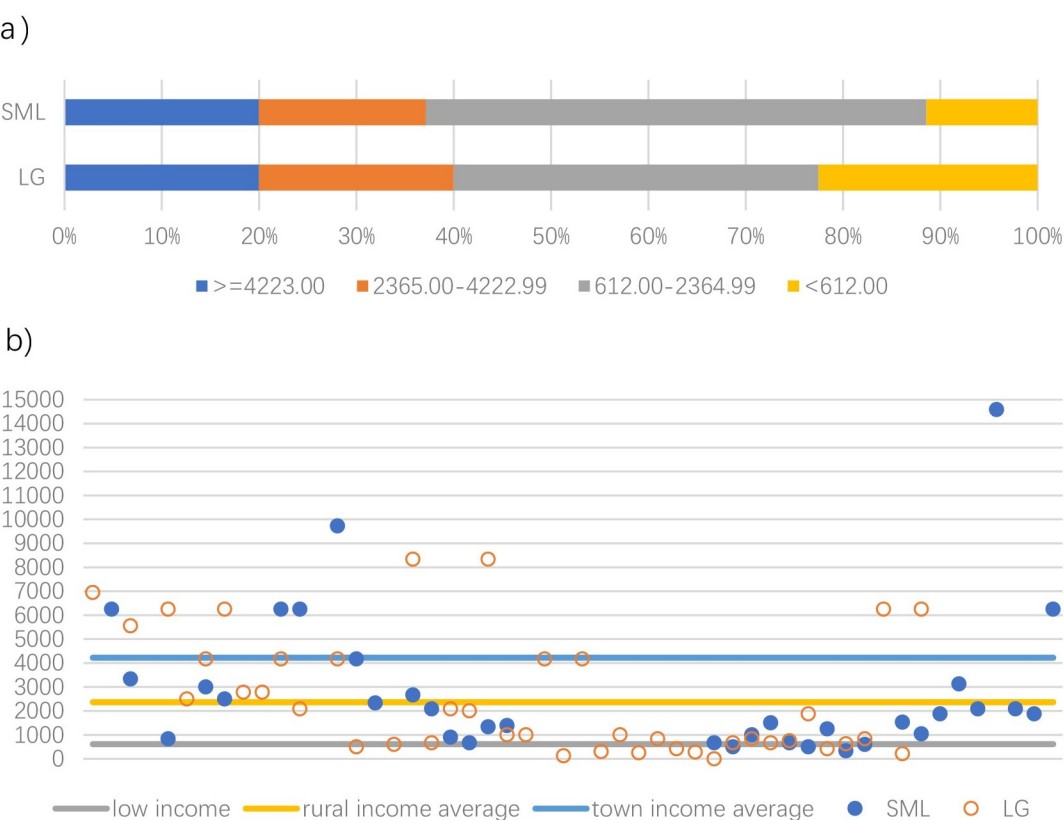

**Fig 5. Income distribution in SML and LG (CNY/per month) a) bar chart; b) scatter chart.**

however, there was no significant deviation in absolute household income between SML and LG. However, the section of the population that lived in abject poverty was, proportionally larger in LG than SML.

## Comparative analysis of local economies and household incomes between SML and LG

Income satisfaction has become the most significant issue in achieving rural management in China [71]. A comparison of the satisfaction of local residents with regards to local economic development is presented in Fig 6 and shows that the advance of tourism can bring both positive and negative influences to the daily lives of local residents in SML. There are three sets of satisfaction data with differences of mean value greater than 0.4, including the fish breeding industry, the dividends that are received from the village company, and the growth of household income over the last 5 years. The village owned company in SML seemed to have made a success of managing local resources, especially with regard to leasing land for the fish breeding industry. SML residents made greater profits from the dividends of the village company than their counterparts in LG. It is therefore not surprising that SML participants were happier with their income over the last five years than residents of LG.

In addition, SML participants had a better experience (mean differences >0.2) of fishing resources, current working status, household income expectations, and with regard to social security. With reference to fishing resources, LG participants were pessimistic as to the state of natural resources to support their pillar industry with 75 per cent of them selecting 3 or below.

a)

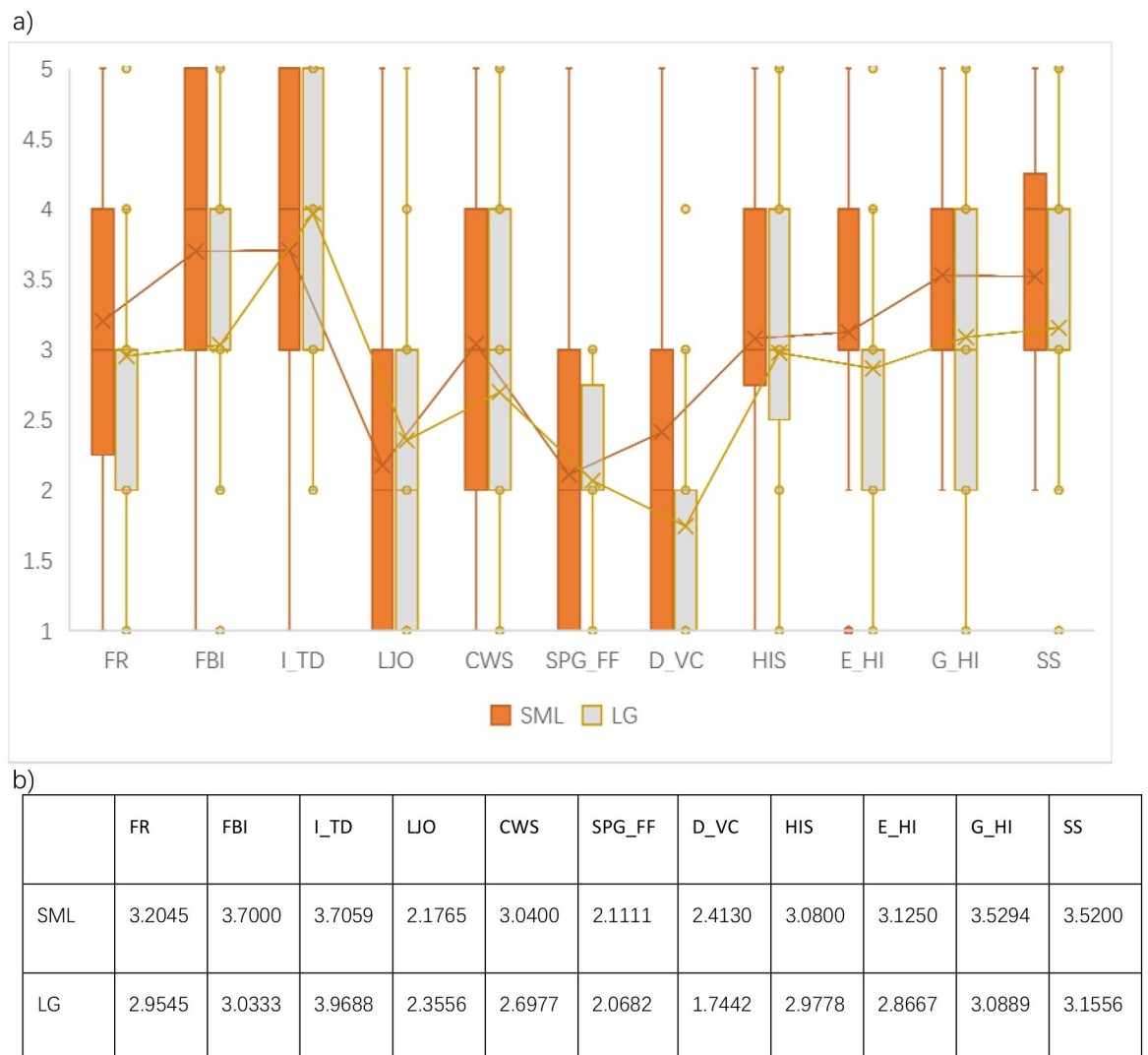

b)

|  | FR | FBI | I_TD | LJO | CWS | SPG_FF | D_VC | HIS | E_HI | G_HI | SS |
|---|---|---|---|---|---|---|---|---|---|---|---|
| SML | 3.2045 | 3.7000 | 3.7059 | 2.1765 | 3.0400 | 2.1111 | 2.4130 | 3.0800 | 3.1250 | 3.5294 | 3.5200 |
| LG | 2.9545 | 3.0333 | 3.9688 | 2.3556 | 2.6977 | 2.0682 | 1.7442 | 2.9778 | 2.8667 | 3.0889 | 3.1556 |

**Fig 6. Descriptive statistics of residents' satisfaction in economic development.** a) boxplot b) mean.

Moreover, the transformation of the industry provided a comfortable working environment and more optimistic career prospects in SML. Half of the SML participants selected 3 and 4 in the question that related to income expectations for 2018, while half of the LG participants selected 2 and 3. Participants in SML showed more confidence in their village's economic development than did their counterparts in LG.

In addition, there was no significant difference (mean difference >0, <0.2) between SML and LG participants with regards to their perceptions relating to changing government subsidies and household income levels. Also, not as same as the labor-intensive fishery industry, tourism could offer limited job opportunities at the initial stage of industrial transformation in SML. Local residents thus could have slightly more negative impression on the local job market created by tourism development. Generally, residents in SML were more satisfied with the local economy than LG except for the impact of tourism development and local job opportunities.

Fig 6 also shows two issues that affect the local economy in LG. More than 50 percent of LG participants selected unsatisfied when questioned as to their view of the government subsidy for fishing and farming. It seems that fishing practitioners in LG are more sensitive to the shifting policy of the subsidy than those in SML. Another issue that can be seen from the data is the condition of the village owned company in LG. The average satisfaction is 1.7442 in this column (less than 2, unsatisfied). 75 percent of LG residents choose (very) unsatisfied. In other words, a majority of the local community does not agree with the contention that company provides them with sufficient monetary benefits.

## Regression analysis of household income satisfaction in both coastal villages

A backward design of multiple regression analysis was employed to discover the significance of twelve predictor variables affecting household income satisfaction (HIS). The predictive variables (see Table 3) were the satisfaction levels of natural resources, local economic development, and respondents' household information.

Table 3 displays the tolerance levels and Variance Inflation Factor (VIF) for the multicollinearity of the predictive variables. By checking these two columns (tolerance < .1, VIF < 10), it was found that there was no reason for the predictive variables to influence each other extensively.

A backward design of multiple regressions is also displayed in Table 3. A correlation between the dependent and predictive variables was generated in Model 6. This shows that six of seven predictive variables were statistically significant at the .05 level, including fishing resources (.010), local job opportunities (.000), expectation of household income in 2018 (.000), growth of household income in the last 5 years (.012), numbers of family members (.014), and numbers of family economic activities (.018).

**Table 3. Coefficients and multicollinearity coefficients a (N = 86) predictive variables.**

| Model | 1 | 2 | 3 | 4 | 5 | 6 |
|---|---|---|---|---|---|---|
| Attribute | B | B | B | B | B | B |
| (Constant) | -0.159b | -0.153b | -0.093b | -0.138b | -0.15b | -0.208b |
| FR | -0.194* | -0.194** | -0.194** | -0.204** | -0.202** | -0.186* |
| FBI | 0.021b | 0.021b | | | | |
| I_TD | 0.001b | | | | | |
| LJO | 0.361*** | 0.36*** | 0.359*** | 0.315*** | 0.306*** | 0.263*** |
| CWS | 0.361b | -0.151b | -0.15b | -0.09b | -0.085b | |
| SPG_FF | -0.137b | -0.137b | -0.141b | -0.061b | | |
| D_VC | 0.13b | 0.129b | 0.133b | | | |
| E_HI | 0.728*** | 0.728*** | 0.725*** | 0.703*** | 0.673*** | 0.651*** |
| G_HI | 0.268* | 0.268* | 0.274** | 0.313** | 0.322** | 0.312** |
| SS | 0.113b | 0.114b | 0.117b | 0.126* | 0.118b | 0.116b |
| N_FM | -0.128b | -0.128b | -0.128b | -0.147* | -0.139* | -0.164* |
| N_FEA | 0.164b | 0.164b | 0.161b | 0.192* | 0.171* | 0.193* |

a. Dependent Variable: HIS.

b. not statistical significance.

*** p< 0.001 is highly significant.

** p< 0.01 is very significant.

* p< 0.05 is significant.

**Table 4. Model summary.**

| Model summary b | | | | |
|---|---|---|---|---|
| Model | R | R Square | Adjusted R Square | Std. Error of the Estimate |
| 6 | .898f | .806 | .781 | .509 |

f. Predictors: (Constant), FR, LJO, E_HI, G_HI, SS, N_FM, N_FEA.

a. Dependent Variable: HIS.

Table 4 provides a summary of model 6 in backward multiple regression analysis. The adjusted R square (.781) indicated that 78.1 per cent of the variables were explained by this model. The purpose of this analysis was to discover the potential factors which impact on local residents' satisfaction with their household income in both SML and LG.

## Discussion: Monetary income and local economy

First, the experiences of rural residents could directly reflect the current performance of the local economy, as well as its historic economic development. This is different from the findings within existent literature about the ability of residents in both rural and urban areas. Sun (2018) claimed that public perception lays emphasis on the short-term development of places because [14], through so doing, members of the public can perceive and observe changes rather than realise the long-term vision of regional development. Moreover, Wang and Mell (2019) argued that it is important for decision-makers to make long-term plans by integrating various short-term or upcoming requirements [25]. However, Chinese planning professionals and politicians may reluctantly undertake a process of reconciling interests between public and other stakeholder groups because the decision-makers have limited confidence in the ability of residents to interpret actual developments diachronically and dynamically.

The findings from data analysis of the results in SML and LG replicate the official data in annual fishery industry report stated by NOFB. Fig 7 presents a set of comparative data (2013 and 2017) based on the development of the fishing industry in Xiangshan. Its gross product decreased by 0.42% in the 5 years from 2013. However, the amount of fishing increased by 5.06% during the same period. Working in fishing could not reward workers with an equivalent income to that which they could attain in other jobs with similar workloads in Xiangshan. Moreover, local residents have other sources of income the YPCI and YPLFI (Fig 7) increased 5.05% and 6.69% per year which is much higher than the increased rate of fishing income (1.48% per year). Thus, the data collected in this study and that contained within annual fishing industry reports explain why local residents showed limited willingness to invest in the fishing industry through the lens of financial and with regard to monetary implications. The data from SML and LG could provide an historic image of industrial development. The data collected from local residents indicated, long-term and short-term, synchronic and diachronic, challenges of industrial development in both SML and LG, including the shortage of active labour force, negative working environment, decreasing fishing resources, and the reduction in supportive industrial policy, which hinders the industrial transformation and the delivery of fishery industry chain. The idea highlighted at this juncture concurs with the view of Fei (2015) [21] that rural lives should be analyzed in a diachronic method which enable planning professionals to focus on in-depth human interactions and historic rural problems rather than the modernist blueprints and ambiguous layout of Rural Revitalization in *developing coastal rural areas and Hangzhou BRA*.

Secondly, household income satisfaction (HIS) is a comprehensive measure of residents' experiences in rural development in coastal areas. It can indicate the social trends of industrial

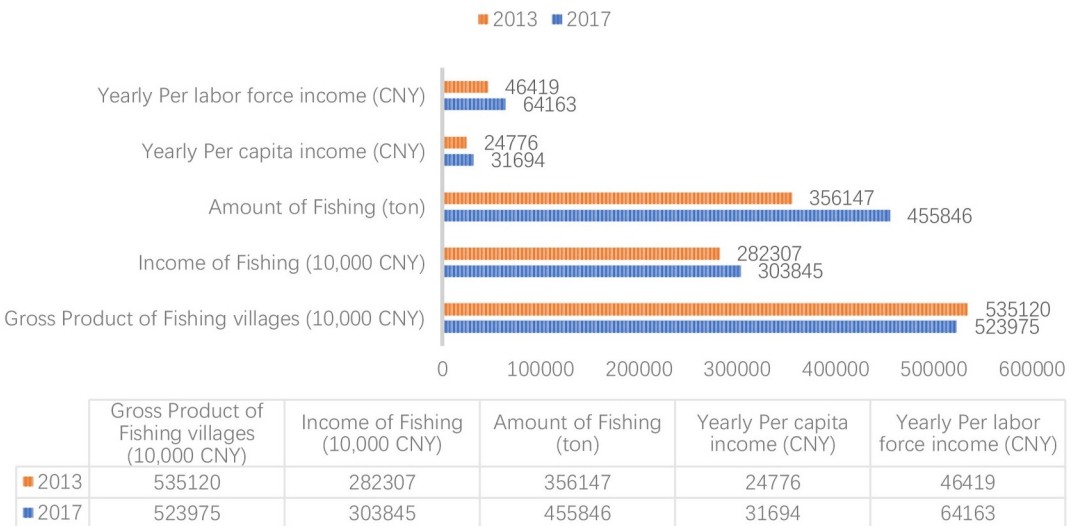

| | Gross Product of Fishing villages (10,000 CNY) | Income of Fishing (10,000 CNY) | Amount of Fishing (ton) | Yearly Per capita income (CNY) | Yearly Per labor force income (CNY) |
|---|---|---|---|---|---|
| 2013 | 535120 | 282307 | 356147 | 24776 | 46419 |
| 2017 | 523975 | 303845 | 455846 | 31694 | 64163 |

**Fig 7. Development of Fishery industry in Xiangshan [72, 73] (edited by first author).**

transformation which could not be understand directly through statistical analysis of economic performance in coastal rural areas. In SML, the residents were more satisfied with their monetary income than their counterparts in LG although the average income of SML residents is lower than LG according to government report (Fig 2). The satisfaction may not, however, directly reflect current absolute income. According to the annual report from NOFB, in 2017, of total 16.88 million CNY fishery family net income in Ningbo, 13.08 million CNY (77.5%) was contributed by fishery industry [72]. This indicates that the tourism industry brought only limited financial benefits to the rural areas, which is also underpinned by the image of annual income (Fig 2). However, the delivery of local tourism impacted on the physical environment and public services, as well as enriching local residents in SML with a regular income. In 2012, more than 80 percent of SML residents became shareholders in a village owned company which conducted land lease transactions and promoted tourism development [58]. SML residents could receive their dividends from land leasing according to a financial disclosure (Fig 8a). Moreover, a majority of residential buildings were re-built and refurbished along with the development of local tourism and homestay industries (Fig 8b). It is also worth noting that facilities and public services, such as car parks, community parks, and bus stops, are more accessible in SML than in LG. The transport facilities gained, however, a certain amount of criticism from LG residents because there are no local no car parks and the bus stop is far away from residential area. This could explain why local residents and decision-makers showed willingness to delivering rural tourism to some extent. In fact, HIS could act as a measure of "thriving business", which indicates social and economic sustainability as a unit in rural industrial development.

Finally, HIS is not an indicator of absolute household income, but rather the expectation and confidence of local residents in the work with which they are engaged. This could further indicate the performance of rural industrial transformation and development. In other words, the residents who have more monetary income may not be satisfied with their income level. This study suggests that HIS is associated with notions of income comparisons. It differs from the classic *Paradox of Happiness* [74] that happiness could not be gained through economic growth whilst, to some extent, supporting the *Relative Income Hypothesis* [75] that people gain happiness and satisfaction by comparing 'their lot' with others. The result also explains the

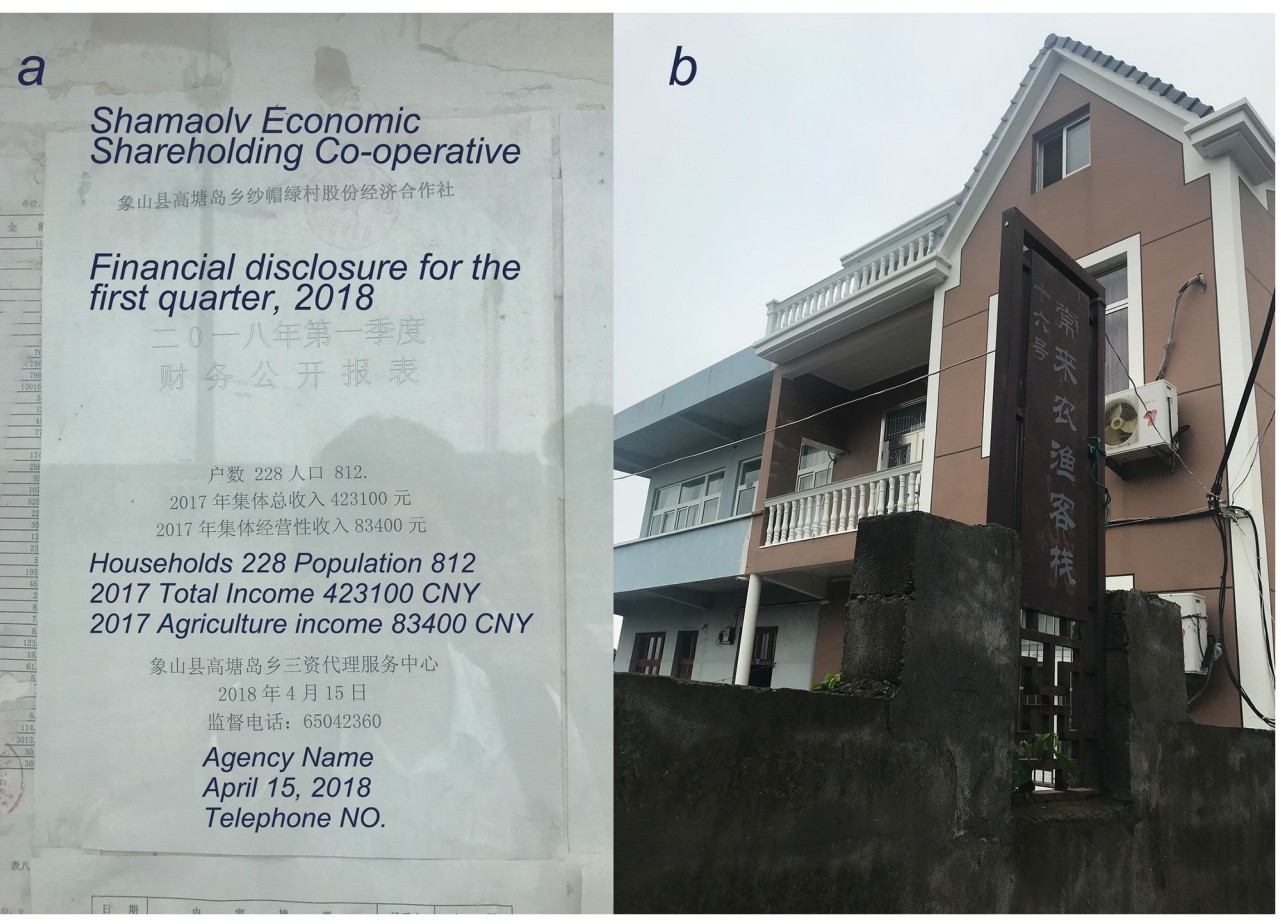

**Fig 8. SML on-site observation.** a) SML financial disclosure for the first quarter, 2018; b) a refurbished residential building for tourism, content on board: No. 16 "Changlai (Welcome)" Fishery Homestay (Source: First author).

work of Sun and Lyu (2020) that cognitive ability of rural elderly residents has strong relationship with their social participation [76].

In addition, plan-makers and decision-makers could evaluate the social equity of industrial transformation through examining HIS in rural development. The social aspect of rural development is relatively more important than economic growth compared with urban areas, which replicates the disparity of prosociality between rural and urban residents in developing counties [77]. The fishery industry brings relatively higher income than tourism at this beginning stage, however, the workload and environment of homestay offices and fishing-boats is entirely different. Further, the agriculture and industry should be delivered gradually to keep its integrity and equality in rural development [78]. The result of data suggests that LG deliver fisheries with traditional manner which met difficulties in fishery resources and labours. This could lead to several in-depth questions on industrialization and transformation of local fisheries, including the delivery of breeding industry, the application of fishing technique and the training system on fishery technician. Low additional value and labour-intensive fishery products become a major source of income which reduces HIS in LG. By comparison, SML residents pay more attention to prospect of engaging tourism development although the tourism projects were delivered gradually and slowly. Their satisfaction on household income can be impacted by the working environment, workload, and regular income brought by the

industrial transformation and "touristization" in coastal rural areas. Consequently, HIS could help to exam the feasibility and integrity of industrial transformation in rural areas, which could reinforce the social equality, social interaction and other issues relating to social sustainability [25].

## Conclusion

This study presented a different account of residents' experiences in economic development in coastal rural areas through an examination of SML and LG in Zhejiang Province. The focus was on discussing the potential for residents' experiences to act as a feasible and valid instrument by which to conduct rural economic analysis, especially to reinforce the social sustainability of rural development. The findings of this study suggest that there exists an alternative process of evaluating economic performance in coastal rural areas in which planning professionals find it difficult to collect data. This finding could contribute to more extensive discussions on rural tourism, hospitality, agriculture, and industrial transformation, and also support the process of decision-making in regional strategic planning in coastal or bay areas. Public participation and stakeholder engagement can substantially underpin the both diachronic and synchronic analysis of local economy in coastal villages.

Over the last five years, China's coastal areas have concentrated much intellectual and financial attention on the development of bay area, such as Hangzhou BRA. A set of visions has been set up by central and local government in respect of promoting urban-rural interaction, tourism, and cultural industries. Such transformations lead to questions arising pertaining to whether rural substance is, and should be, prepared to underpin regional vision. Moreover, whether the status of rural economy, or "thriving business", can be evaluated comprehensively through the lens of residents' experiences is a concept that has yet to be fully addressed. Existing literature suggests that public value can contribute to the assessment of physical and cultural environment [38]. However, there have been limited studies undertaken which support the idea that rural residents can contribute to the in-depth and diachronic analysis of local economies. Moreover, a majority of existing studies represent normal rural areas which are distinct from the coastal areas which have attracted a growing number of investments under the auspices of national and regional strategic planning. In addition, the culture and interpersonal relationships in coastal rural areas differ from those of conventional Chinese rural villages. These questions pertaining to the research lead one to need to assess which aspects of economic development can be better understood, analysed, and planned through residents' experiences, and to question what indicators could best represent residents' economic experiences. One of the current challenges in engaging the public and local community lies in the limited confidence of practitioners on the validity of public perceptions, especially in coastal rural areas where local residents have, generally, only a limited education background.

We conclude that the small number of samples addressed within this paper is nevertheless important and a valid and alternative means by which to analyse the economic performance and history as a whole in coastal rural areas within Chinese context. HIS could act as an indicator of social and economic sustainability in rural development. The result of residents' experiences replicates, but also helps to explain, the official data in annual industrial reports. This, *ceteris paribus*, can allow practitioners to conduct rural economic studies by investigating, comparing, analysing, integrating, and synthesising perceptions of local residents in coastal rural areas. Such an approach also give rise to research opportunities that focus on urban-rural interaction, social inclusion, and social inequality; all of which are already gaining growing attention from practitioners and scholars in the field of planning. Moreover, this research

enables a process of plan-making on rural economic development by integrating perceptions from diachronic public participation and synchronic observation on physical environment.

This study empirically highlights where residents' experiences can be adopted, where new research on residents' experiences is needed, and how existent gaps in understanding can be bridged through research. The findings enable researchers in the field to conduct their work with a blend of methods and data resources, but also with an in-depth understanding of the socio-economic context of coastal rural areas in China. Moreover, practitioners and plan-makers are able to develop scenarios based on the identified weakness and challenges in the actual development of coastal villages.

Finally, it is noted that this study gained insight from the contest discourse of communicative turn in spatial planning to debate over the viewpoint that rural residents' perceptions are diachronic, comprehensive, and informative. It particularly highlighted the role of residents' experiences in the discussion of mid-term and long-term economic development in coastal rural areas which are deemed of importance under the national policy of Rural Revitalization. This study also provides a pilot case study to encourage theoretical and practical researches on identifying, understanding, and articulating the in-depth rationalities that lie behind residential experiences and substantial realities in economically changing and transforming coastal villages. The integration of public perspectives into planning process could also be used as empirical evidences for studies on Collaborative Planning. Further studies could be conducted to explore and elaborate the potential relationship between residential experiences and evaluation of regional economic planning and good governance.

## Supporting information

**S1 File. Questionnaire in English.**
(DOCX)

**S2 File. Questionnaire in original language.**
(DOCX)

## Acknowledgments

We thank Dr Bertie Dockerill for his professional proofreading on first draft. We also thank Hui Yang, Songli Shi, Xi Shen, Yichen Yu, and others in The Rural Development and Ecological Technology Research Institute for their help during data collection.

## Author Contributions

**Conceptualization:** Xinkai Wang.

**Data curation:** Yao Wang.

**Formal analysis:** Xinkai Wang, Yao Wang.

**Funding acquisition:** Xinkai Wang, Zhirong Wang, Qianqian Zhang.

**Investigation:** Yao Wang, Qianqian Zhang, Tengyue Zhang.

**Methodology:** Xinkai Wang.

**Software:** Xinkai Wang, Jia Yao.

**Supervision:** Zhirong Wang.

**Writing – original draft:** Xinkai Wang.

**Writing – review & editing:** Xinkai Wang, Zhirong Wang, Jia Yao.

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
