## [Decision Letter · Decision Letter 0]

4 Jun 2020

PONE-D-20-12170

How residents’ experiences indicate changing economic environment of coastal villages: Evidence from the Greater Hangzhou Bay Rim Area

PLOS ONE

Dear Dr. wang,

Thank you for submitting your manuscript to PLOS ONE. After careful consideration, we feel that it has merit but does not fully meet PLOS ONE’s publication criteria as it currently stands. Therefore, we invite you to submit a revised version of the manuscript that addresses the points raised during the review process.

We look forward to receiving your revised manuscript.

Kind regards,

Bing Xue, Ph.D.

Academic Editor

PLOS ONE

Journal Requirements:

3. Please ensure that you include a title page within your main document. We do appreciate that you have a title page document uploaded as a separate file, however, as per our author guidelines (http://journals.plos.org/plosone/s/submission-guidelines#loc-title-page) we do require this to be part of the manuscript file itself and not uploaded separately.

4. Please include your tables as part of your main manuscript and remove the individual files. Please note that supplementary tables (should remain/ be uploaded) as separate "supporting information" files.

5. Please ensure that you refer to Figure 2 in your text as, if accepted, production will need this reference to link the reader to the figure.

6. Please upload a copy of Figure 7, to which you refer in your text on page xx. If the figure is no longer to be included as part of the submission please remove all reference to it within the text. We note there are two figures currently labelled as figure 1 in the figure files so please make sure that the figure labels are carefully checked and corrected as needed.

7. We note that Figure 1 and 6 in your submission contain map images which may be copyrighted. All PLOS content is published under the Creative Commons Attribution License (CC BY 4.0), which means that the manuscript, images, and Supporting Information files will be freely available online, and any third party is permitted to access, download, copy, distribute, and use these materials in any way, even commercially, with proper attribution. For these reasons, we cannot publish previously copyrighted maps or satellite images created using proprietary data, such as Google software (Google Maps, Street View, and Earth). For more information, see our copyright guidelines: http://journals.plos.org/plosone/s/licenses-and-copyright.

7.1.    You may seek permission from the original copyright holder of Figure 1 and 6 to publish the content specifically under the CC BY 4.0 license.

7.2.    If you are unable to obtain permission from the original copyright holder to publish these figures under the CC BY 4.0 license or if the copyright holder’s requirements are incompatible with the CC BY 4.0 license, please either i) remove the figure or ii) supply a replacement figure that complies with the CC BY 4.0 license. Please check copyright information on all replacement figures and update the figure caption with source information. If applicable, please specify in the figure caption text when a figure is similar but not identical to the original image and is therefore for illustrative purposes only.

8. We note that Figure 5 includes an image of a participant in the study. 

Reviewers' comments:

Reviewer's Responses to Questions

**Comments to the Author**

1. Is the manuscript technically sound, and do the data support the conclusions?

Reviewer #1: No

2. Has the statistical analysis been performed appropriately and rigorously? 

Reviewer #1: No

3. Have the authors made all data underlying the findings in their manuscript fully available?

Reviewer #1: Yes

4. Is the manuscript presented in an intelligible fashion and written in standard English?

Reviewer #1: Yes

5. Review Comments to the Author

Reviewer #1: 1. The contribution to the body of scientific knowledge is poor. I am not convinced that it saying something particularly new about the rural planning and the local economic development.

2. Unsophisticated methodology was considered and typical results were concluded.

3. Lack of justifications and information of the selected study area (two coastal villages).

4. Lack of in-depth analysis of the rural development, rural tourism, agriculture and industrial transformation.

5. The magnitude of HI effect on resident’s experiences and changing economic environment which seems to be the main focus of the paper is not clear and unjustified.

6. The plausibility and relevancy of the results for changing economic environment of coastal villages.

7. Economic status and growth in china (particularly in eastern and central regions) seems to be the main catalyst of both development and transportation infrastructure growth, how this variable might affect the result of this study.

8. The main scientific messages and findings are missing.

6. PLOS authors have the option to publish the peer review history of their article (what does this mean?). If published, this will include your full peer review and any attached files.

Reviewer #1: No

---

## [Author Response · Author response to Decision Letter 0]

15 Jul 2020

Dear reviewer,

Thank you for the time and effort that you have put into reviewing the previous version of the manuscript. Your suggestions enabled us to polish the work. Based on the instructions provided in your letter, we have revised the manuscript accordingly. Details of revision has been provided in the rebuttal letter. Thank you again for your precious time and work.

Sincerely,

Authors

---

## [Decision Letter · Decision Letter 1]

18 Aug 2020

PONE-D-20-12170R1

Integration of residents' experiences into economic planning process of coastal villages: Evidence from the Greater Hangzhou Bay Rim Area

PLOS ONE

Dear Dr. Wang,

Thank you for submitting your manuscript to PLOS ONE. After careful consideration, we feel that it has merit but does not fully meet PLOS ONE’s publication criteria as it currently stands. Therefore, we invite you to submit a revised version of the manuscript that addresses the points raised during the review process.

We look forward to receiving your revised manuscript.

Kind regards,

Bing Xue, Ph.D.

Academic Editor

PLOS ONE

Reviewers' comments:

Reviewer's Responses to Questions

**Comments to the Author**

1. If the authors have adequately addressed your comments raised in a previous round of review and you feel that this manuscript is now acceptable for publication, you may indicate that here to bypass the “Comments to the Author” section, enter your conflict of interest statement in the “Confidential to Editor” section, and submit your "Accept" recommendation.

Reviewer #1: All comments have been addressed

Reviewer #2: All comments have been addressed

2. Is the manuscript technically sound, and do the data support the conclusions?

Reviewer #1: Yes

Reviewer #2: Partly

3. Has the statistical analysis been performed appropriately and rigorously? 

Reviewer #1: Yes

Reviewer #2: (No Response)

4. Have the authors made all data underlying the findings in their manuscript fully available?

Reviewer #1: Yes

Reviewer #2: Yes

5. Is the manuscript presented in an intelligible fashion and written in standard English?

Reviewer #1: Yes

Reviewer #2: Yes

6. Review Comments to the Author

Reviewer #1: (No Response)

Reviewer #2: I just wondering why the discussion specifically focus on monetary income and local economy, while this paper has a potential conclusion related to the context of coastal villages, while economy side will be supported by other factors such as geographical uniqueness as well as environmental and social facets.

I am also concern about the conclusion which has been revised by the author. It is interesting if the role of the community in terms of economic planning process can be connected to criticize existing planning policy or even global issue of participatory/collaborative planning, innovative governance, and special interest tourism.

7. PLOS authors have the option to publish the peer review history of their article (what does this mean?). If published, this will include your full peer review and any attached files.

Reviewer #1: No

Reviewer #2: No

---

## [Author Response · Author response to Decision Letter 1]

20 Aug 2020

Dear Reviewers,

Thanks very much for taking your time to review the revised manuscript. We appreciate all your comments and suggestions. Please find our itemized responses in the re-submitted files.

---

## [Decision Letter · Decision Letter 2]

21 Sep 2020

Integration of residents' experiences into economic planning process of coastal villages: Evidence from the Greater Hangzhou Bay Rim Area

PONE-D-20-12170R2

Dear Dr. Wang,

We’re pleased to inform you that your manuscript has been judged scientifically suitable for publication and will be formally accepted for publication once it meets all outstanding technical requirements.

Kind regards,

Bing Xue, Ph.D.

Academic Editor

PLOS ONE

Additional Editor Comments (optional):

Reviewers' comments:

Reviewer's Responses to Questions

**Comments to the Author**

1. If the authors have adequately addressed your comments raised in a previous round of review and you feel that this manuscript is now acceptable for publication, you may indicate that here to bypass the “Comments to the Author” section, enter your conflict of interest statement in the “Confidential to Editor” section, and submit your "Accept" recommendation.

Reviewer #1: (No Response)

2. Is the manuscript technically sound, and do the data support the conclusions?

Reviewer #1: Yes

3. Has the statistical analysis been performed appropriately and rigorously? 

Reviewer #1: Yes

4. Have the authors made all data underlying the findings in their manuscript fully available?

Reviewer #1: Yes

5. Is the manuscript presented in an intelligible fashion and written in standard English?

Reviewer #1: Yes

6. Review Comments to the Author

Reviewer #1: (No Response)

7. PLOS authors have the option to publish the peer review history of their article (what does this mean?). If published, this will include your full peer review and any attached files.

Reviewer #1: No

---

## [Editor Report · Acceptance letter]

29 Sep 2020

PONE-D-20-12170R2 

Integration of residents’ experiences into economic planning process of coastal villages: Evidence from the Greater Hangzhou Bay Rim Area 

Dear Dr. Wang:

I'm pleased to inform you that your manuscript has been deemed suitable for publication in PLOS ONE. Congratulations! Your manuscript is now with our production department. 

Kind regards, 

on behalf of

Professor Bing Xue 

Academic Editor

PLOS ONE